# Angiogenesis Still Plays a Crucial Role in Human Melanoma Progression

**DOI:** 10.3390/cancers16101794

**Published:** 2024-05-08

**Authors:** Gerardo Cazzato, Giuseppe Ingravallo, Domenico Ribatti

**Affiliations:** 1Section of Molecular Pathology, Department of Precision and Regenerative Medicine and Ionian Area (DiMePRe-J), University of Bari “Aldo Moro”, 70124 Bari, Italy; giuseppe.ingravallo@uniba.it; 2Department of Translational Biomedicine and Neuroscience, University of Bari Medical School, 70124 Bari, Italy; domenico.ribatti@uniba.it

**Keywords:** angiogenesis, anti-angiogenesis, melanoma, metastasis, tumor progression

## Abstract

**Simple Summary:**

Despite the advancement of molecular biology and Next-Generation Sequencing (NGS), the study of the tumor microenvironment continues to have paradigmatic importance in research applied to malignant melanoma, and neoangiogenesis continues to exert a significant influence. In this review, we discuss the most updated knowledge about neoangiogenesis in melanoma, address the role of the TME in progression, and outline future therapeutic perspectives.

**Abstract:**

Angiogenesis plays a pivotal role in tumor progression, particularly in melanoma, the deadliest form of skin cancer. This review synthesizes current knowledge on the intricate interplay between angiogenesis and tumor microenvironment (TME) in melanoma progression. Pro-angiogenic factors, including VEGF, PlGF, FGF-2, IL-8, Ang, TGF-β, PDGF, integrins, MMPs, and PAF, modulate angiogenesis and contribute to melanoma metastasis. Additionally, cells within the TME, such as cancer-associated fibroblasts, mast cells, and melanoma-associated macrophages, influence tumor angiogenesis and progression. Anti-angiogenic therapies, while showing promise, face challenges such as drug resistance and tumor-induced activation of alternative angiogenic pathways. Rational combinations of anti-angiogenic agents and immunotherapies are being explored to overcome resistance. Biomarker identification for treatment response remains crucial for personalized therapies. This review highlights the complexity of angiogenesis in melanoma and underscores the need for innovative therapeutic approaches tailored to the dynamic TME.

## 1. Introduction

Angiogenesis is a crucial event in tumor progression and is necessary not only for the transport of O_2_ and nutrition supply to cancerous cells but also for the removal of CO_2_ and waste products [1,2]. It has been known that angiogenesis is controlled by a delicate balance between molecules that have positive and, respectively, negative regulatory activity, which constitutes a crucial and very complex point also in the regulation of cancer metabolism [3]. Angiogenesis is divided into two phases (so-called angiogenic switch): (1) avascular phase, in which metabolites and catabolites are transferred by simple diffusion through the surrounding tissue, and (2) vascular phase, in which there is the formation of new blood vessels starting from the capillaries and from the post-capillary venules, and the tumor acquires the characteristics of progression such as local invasion and hematogenous metastasis [4].

Melanoma represents the most lethal form of skin cancer, whose incidence and prevalence rates have increased over the last decade [5], with decreased mortality rates both for improved early diagnosis and the effect of immune checkpoint inhibitors in therapy. From a histopathological point of view, different histotypes are identified, among which the most common is represented by the Superficial Spreading Type (SSM), in which there is a phase of horizontal growth (radial growth phase) and then a phase of vertical growth, which can be of the tumorigenic (mass-forming) or non-tumorigenic type [6,7]. With the advent of molecular biology and Next-Generation Sequencing (NGS), there has been an increase in new acquisitions regarding potential mutations affecting cutaneous melanoma [8], but, on the other hand, the concept of tumor microenvironment (TME) has continued to exert its influence, with increasingly sophisticated studies seeking to characterize individual “cellular actors” (i.e., single-cell sequencing) [9]. In this context, angiogenesis plays a role of paramount importance, as it can modify the TME and exert an influence on the probability of metastasis [10]. Furthermore, it has previously been demonstrated that there is a directly proportional correlation between intratumoral microvessel density (MVD), vascular proliferation index (VPI), overall survival (OS), adverse prognosis, Breslow thickness, and risk of recurrence [11,12,13,14,15], as well as the ability of melanoma cells to acquire the expression of the laminin receptor, so as to adhere more tightly to the vessel wall [16], favoring and permitting tumor cell extravasation and metastases [17]. It is also important to consider that it has been demonstrated that a higher lymphatic vessel density (LVD) and lymphatic vessel area (LVA) were significantly associated with SLN metastasis; high intra-tumoral and peritumoral LVD were also significantly correlated with shorter overall survival in patients with melanoma [18].

Secreted factors can be induced by activation of different receptors; for example, activation of nAChRs can cause the secretion of VEGF, but receptors can also form a functional network. Thus, angiogenic factors can also mediate the progression of melanomas and vice versa [19].

In this review, we provide the most up-to-date knowledge on angiogenesis in melanoma, discuss potential uncertainties, and try to answer new questions about the relationship between neoangiogenesis and TME.

## 2. Pro-Angiogenic Factors

### 2.1. Vascular Endothelial Growth Factor (VEGF)

VEGF/vascular permeability factor (VPF) has been identified as a multifunctional peptide involved in endothelial cell proliferation, migration, and survival in both healthy and pathological conditions [20]. Placental growth factor (PlGF), VEGF-A (often referred to as VEGF), VEGF-B, VEGF-C, VEGF-D, and VEGF-E constitute the human VEGF family, and all these isoforms share the same tyrosine kinase receptors [21]. It has been demonstrated that VEGF and its receptor play the most important function in angiogenesis, even though several endogenous factors, including fibroblast growth factor-2 (FGF-2); hypoxia-inducible factor-1 alpha and beta (HIF-1α/β); tumor necrosis factor alpha (TNF-α); transforming growth factor alpha and beta (TGF-α and β); interleukin-8 (IL-8); angiopoietin-1 and-2 (Ang-1 and Ang-2); platelet-derived growth factor (PDGF); and epidermal growth factor (EGF), play a role as angiogenesis mediators [22]. VEGF-A secreted by melanoma binds with high-affinity the receptors VEGFR-1 and VEGFR-2, playing a pivotal role in angiogenesis in melanoma [23,24]. In primary melanoma, there is a positive correlation between a higher tumor thickness (>3.6 mm) and factors such as increased vessel diameter, a high number of connections between intraluminal tissue folds and the opposing vascular wall, increased vessel density, and significant VEGF-A positivity of tumor cells [25]. Overexpression of isoforms VEGF121 and VEGF165 induced tumor growth in human melanoma xenografts, but overexpression of VEGF189 had no such effect [26], while VEGF-A-expressing tumor-infiltrating inflammatory cells are linked to the up-regulation of VEGF-A expression in the setting of metastatic melanoma [27]. Overexpression of melanotransferrin, an angiogenic protein both in vivo and in vitro, has been linked to the expression of VEGF-A in human melanoma [28]. Other studies confirmed a positive correlation between increased VEGF-A immune expression and the transition from the radial phase to the vertical phase of melanoma or even from primary to metastatic melanoma [29,30,31,32,33].

VEGF-C and VEGF-D act as lymphangiogenic growth factors signaling via VEGFR-2 and VEGFR-3, and the lack of the VEGF-C in mice results in compromised development of the lymphatic vasculature [34,35]. VEGF-C is expressed in primary and metastatic melanoma and is associated with high intratumoral lymphatic density [36,37,38]. VEGF-C, VEGF-D, and their receptor VEGFR-3 are crucial for lymphangiogenesis and the lymphatic dissemination of cancer cells, even if an inverse relationship has been demonstrated between the level of these markers and lymph node metastasis [18]. No discernible variation in tumor cells’ expression of VEGF-C between metastatic and non-metastatic melanomas has been found [39]. VEGF-C expression was greater in melanomas with sentinel lymph node positivity than in those without [40]. As tumor invasion and migration are also regulated by other mediators, VEGF-C, VEGF-D, and VEGFR-3 should not be considered the sole variables implicated in lymphatic dissemination and metastasis [41,42,43].

### 2.2. Placental Growth Factor (PlGF)

PlGF is recognized as a pro-angiogenic growth factor, binding neuropilin-1 and -2 receptors expressed on endothelial cells [44], as well as VEGFR-1 and forming heterodimers with VEGF-A [45,46]. Transgenic mice overexpressing PlGF developed a greater density of blood vessels, followed by a tumor growth rate five times higher than controls, a greater number and size of lung melanoma metastases, and a greater entity of tumor cell invasion of the extracellular matrix (ECM) [47]. More recently, a study investigated the role of PlGF NRP-1 in promoting melanoma aggressiveness independently of VEGFR-1. The researchers used melanoma cell clones M14-N, expressing NRP-1 but lacking VEGFR-1, and M14-C, devoid of both receptors. M14-N cells displayed an invasive phenotype and vasculogenic mimicry, while M14-C cells had minimal invasive capacity. The results showed that M14-N cells secreted higher levels of PlGF compared to M14-C cells, and PlGF played a role in ECM invasion and vasculogenic mimicry of M14-N cells. Furthermore, neutralizing antibodies against PlGF reversed ECM invasion and reduced tubule-like structure formation. Additionally, M14-N cells migrated in response to PlGF, and chemotaxis was inhibited by anti-NRP-1 antibodies, indicating that PlGF directly activates NRP-1 in the absence of VEGFR-1. Finally, plasma PlGF levels were compared between metastatic melanoma patients and healthy donors. Melanoma patients exhibited a 20-fold increase in plasma PlGF levels, and interestingly, a bevacizumab-containing chemotherapy regimen led to a reduction in VEGF-A levels and a further increase in PlGF levels [34]. A more recent study focuses on investigating the role of PlGF and its interaction with VEGFR-1 in tumor-associated angiogenesis and melanoma invasion of the extracellular matrix (ECM). The researchers explored the effects of an anti-VEGFR-1 monoclonal antibody (D16F7 mAb) on melanoma spread. This antibody inhibits VEGF-A or PlGF-mediated signal transduction without disrupting ligand interaction, thereby preserving the function of soluble VEGFR-1 (sVEGFR-1) as a decoy for proangiogenic factors.

The study aims to determine whether D16F7 mAb hampers melanoma spread through various in vitro and in vivo analyses. In vitro experiments involve assessing melanoma cell adhesion to sVEGFR-1, ECM invasion, and transmigration through an endothelial cell monolayer. In vivo evaluations include examining tumor infiltrative potential in a syngeneic murine model.

The results demonstrate that D16F7 mAb significantly inhibits melanoma adhesion to sVEGFR-1, ECM invasion, and transmigration induced by PlGF. Treatment with the antibody not only inhibits tumor growth but also leads to a significant reduction in bone infiltration associated with a decrease in PlGF-positive melanoma cells in mice. Additionally, D16F7 mAb reduces PlGF production by melanoma cells. In conclusion, this study suggested that blocking PlGF/VEGFR-1 signaling with D16F7 mAb represents a promising strategy to counteract the metastatic potential of melanoma by inhibiting tumor-associated angiogenesis and invasion [35].

### 2.3. Fibroblast Growth Factor-2 (FGF-2)

FGF-2 is a pro-angiogenic cytokine of the FGFs family that comprises 22 structurally related proteins that exert their effects by binding to and activating specific cell surface receptors implicated in tumorigenesis [48]. Numerous studies have demonstrated aberrant expression of FGF-2 and FGF18, as well as increased FGFR1 and FGFR3 levels, in primitive melanoma samples compared to normal skin tissue [49], and a significant relationship between high microvascular density and expression of FGFR4 has been demonstrated [50]. Furthermore, FGF-2 and endostatin are potentially useful biomarkers for early detection during melanoma progression [51], while VEGF-dependent neovascularization in a mouse melanoma model induced by FGF-2 has been shown [52]. FGF-2 may be induced by the activity of matrix metalloproteinases (MMPs) released by melanoma cells [27], and a significant correlation between the percentage of FGF-expressing tumor cells, the number of mast cells, and melanoma progression has been evidenced [53]. 

### 2.4. Interleukin-8 (IL-8)

IL-8 promotes the proliferation and survival of endothelial cells and cancer cells, favoring the modulation of TME with the migration of endothelial cells, changes in vascular permeability, and enhanced actin stress fiber formation [54]. IL-8 exerts pro-angiogenic functions by binding to its receptors, CXCR1 and CXCR2 [55]. Normal epidermis and benign melanocytic lesions express very low quantities of IL-8, while in patients with melanoma, high IL-8 serum levels have been demonstrated, correlated with both an advanced disease stage and overall survival [54]. IL-8 binds its receptors CXCR1 and CXCR2 and exerts pro-angiogenic functions, as demonstrated by Singh et al. [55]. In human melanoma xenografts, a relationship between hypoxia, IL-8, angiogenesis, and metastasis has been established, and neutralizing antibodies against IL-8 also decreased the incidence of metastases and vascular density [56]. Tumor production of IL-8 drives melanoma cell development and enhances tumor cell movement [57].

### 2.5. Angiopoietin (ANG)

Ang family consists of four members, namely Ang-1, Ang-2, Ang-3, and Ang-4. Among these, Ang-1 and Ang-2 are the most extensively studied [58]. Ang-1 is mainly produced by pericytes and smooth muscle cells surrounding blood vessels. It binds to the Tie-2 receptor on endothelial cells, leading to stabilization and maturation of blood vessels, promoting vessel quiescence, and inhibiting vascular permeability [59]. Ang-2 is released by endothelial cells themselves, particularly during vascular remodeling and inflammation, and acts as a context-dependent antagonist or agonist of Tie-2 signaling, and, in the presence of low levels of VEGF, destabilizes blood vessels and promotes angiogenesis. However, in the presence of high levels of VEGF, Ang-2 acts as a Tie-2 agonist and promotes vessel stabilization [60]. Both Ang-1 and Ang-2 are expressed in melanoma tumors; however, the balance between Ang-1 and Ang-2 expression is altered in melanoma, favoring angiogenesis, and high levels of Ang-2 relative to Ang-1 are associated with increased tumor vascularity, tumor progression, and poor prognosis [61,62].

### 2.6. Transforming Growth Factor-Beta (TGF-β)

TGF-β is a multifunctional cytokine that plays crucial roles in various cellular processes, including cell growth, differentiation, migration, apoptosis, and immune regulation [63]. In melanoma, TGF-β signaling pathway dysregulation has been implicated in tumor progression, invasion, and metastasis, including angiogenesis [63]. TGF-β exerts both pro-angiogenic and anti-angiogenic effects in the context of melanoma; its role in angiogenesis is influenced by various factors such as tumor stage, microenvironment, and interactions with other signaling pathways [63]. TGF-β promotes angiogenesis in melanoma through several mechanisms: (1) upregulation of the expression of other angiogenic factors such as VEGF, FGF-2, and Ang-1. (2) Recruitment of endothelial progenitor cells (EPCs) and incorporation of EPCs into neovessels, facilitating angiogenesis [64]. (3) Induction of endothelial–mesenchymal transition (EndMT), leading to the acquisition of mesenchymal characteristics and enhanced angiogenic potential [65]. TGF-β can also exert anti-angiogenic effects in melanoma through the following: (1) inhibition of endothelial cell proliferation by inducing cell cycle arrest and apoptosis [66]; (2) regulation of ECM remodeling influencing the formation and stability of blood vessels; (3) and crosstalk with other signaling pathways involved in angiogenesis, such as Notch and Wnt pathways, modulating endothelial cell behavior and vessel formation [67].

### 2.7. Platelet-Derived Growth Factor (PDGF)

PDGF comprises four family members (PDGF-A, -B, -C, and -D) and its receptors, PDGFR (PDGFR-A and PDGFR-B), overexpressed in melanoma cells [68]. PDGF acts as a potent mitogen for melanoma cells, stimulating their proliferation by activating downstream signaling pathways such as the MAPK/ERK and the PI3K/Akt pathway. PDGF signaling promotes cell cycle progression and inhibits apoptosis, thereby enhancing the survival and growth of melanoma cells [69]. Human melanoma xenografts produced from B16 cells transfected with PDGF-BB showed vasculature with a greater pericyte coverage [70]. Furthermore, MRI analysis demonstrated that PDGF-BB caused a decrease in vessel diameter and an increase in the degree of tumor blood vascular perfusion [70]. When B16 melanoma cells were transplanted into tumors in mice expressing an active PDGF receptor beta, the average vessel surface and total vessel area were greater than in wild-type mice [71]. Melanoma cells overexpressing PDGFR-A develop tumors that are smaller in weight and are less angiogenic when compared to controls [72]. Moreover, PDGFR-A strongly inhibits melanoma growth both in vitro and in vivo [72].

## 3. Integrin Signaling and Extracellular Matrix Enzymes

### 3.1. Integrins

Integrins are heterodimeric transmembrane cell surface receptors that mediate cell–cell and cell–ECM interactions, playing important roles in cell adhesion, migration, proliferation, and survival [73]. Integrins are composed of α and β subunits, and the engagement with ECM ligands triggers intracellular signaling cascades in melanoma, including the following: (1) Focal adhesion kinase (FAK) pathway, in which integrin-mediated adhesion activates FAK, which in turn phosphorylates downstream effectors such as Src kinase, PI3K/Akt, and ERK1/2, promoting cell migration, survival, and proliferation [74]. (2) Rho GTPase pathway in which integrins regulate Rho GTPases, which control actin cytoskeleton dynamics, cell polarity, and migration [75]. (3) MAPK/ERK pathway, promoting cell proliferation, survival, and invasion [76]. (4) PI3K/Akt/mTOR pathway, which regulates cell growth, metabolism, and survival of melanoma cells [77]. The shift from primary to metastatic melanoma has been linked to overexpression of integrins αvβ3, αvβ5, α2β1, and α5β1 [78] and, consequently, overexpression of integrins increases the propensity of melanoma cells to invade by stimulating MMP-2 and MMP-7 [79].

### 3.2. Matrix Metalloproteinases (MMPs)

MMPs are a family of zinc-dependent endopeptidase enzymes consisting of more than 20 members in humans, classified based on their substrate specificity and domain structure, which are crucial for wound healing, tissue remodeling, and angiogenesis [80]. They are secreted as inactive zymogens (pro-MMPs) and require activation by proteolytic cleavage of the pro-domain to become active, and MMPs can degrade various components of the ECM, including collagens, gelatin, elastin, fibronectin, and laminin, thereby facilitating cell migration, invasion, and angiogenesis [81]. The dysregulation of expression and activity of MMPs have been implicated in the progression and metastasis of melanoma [82]; indeed, melanoma cells and tumor-associated stromal cells produce and secrete MMP-2, MMP-9, MMP-1, and MMP-3, contributing to ECM remodeling and tumor progression and various factors, including growth factors, cytokines, and hypoxia, can stimulate MMP expression and activity in melanoma cells through signaling pathways such as MAPK/ERK, PI3K/Akt, and NF-κB [83]. Overexpression of MMPs has been associated with reduced survival rate, overexpression of Bcl-2, and increased microvascular density [82,83,84], while MMP-2 expression was highly correlated with metastatic spread and low survival rates [82]. The values of MMPs have been evaluated in tissue samples from primary nodular melanoma (NM) and dysplastic nevus using immunohistochemical staining [84]. The study also investigated the association between MMP expression, clinicopathological variables, BRAF V600 mutation status, and overall survival, and authors reported that primary NM tumors exhibited significantly higher expression levels of MMP-1, MMP-2, and MMP-13 compared to dysplastic nevi, and higher MMP-1 expression was associated with worse clinical outcomes, suggesting its potential as an independent prognostic marker. Furthermore, patients with BRAF V600 mutation did not show significant differences in overall survival compared to BRAF wild-type patients. Interestingly, BRAF V600 mutated NM samples exhibited lower expression levels of MMP-1 and MMP-13 compared to BRAF wild-type samples. In terms of survival analysis, features such as Clark level IV and V, Breslow thickness >4 mm, and high MMP-1 protein expression (>30%) were identified as prognostic factors for shorter overall survival in patients with NM. MMP-9 expression was found exclusively in the horizontal growth phase of melanoma, not in the vertical phase, demonstrating that MMP-9 expression is a precursor to melanoma development [85]. The balance between MMPs and their inhibitors (TIMPs) ultimately determines the progression of melanoma, as many investigations have shown using either animal models or cell lines. Overexpression of TIMP-1, -2, and -3 lowers tumor neovascularization and greatly decreases melanoma tumor cell invasion, migration, and metastasis [86].

### 3.3. Platelet-Activating Factor (PAF)

PAF (also called AGEPC) stimulates the proliferation of melanoma cells by activating intracellular signaling pathways such as the MAPK/ERK pathway and the PI3K/Akt pathway, regulating cell cycle progression and promoting cell growth [87]. PAF can inhibit apoptosis in melanoma cells by modulating the expression of anti-apoptotic proteins such as Bcl-2 and Bcl-xL, and by suppressing the activation of pro-apoptotic proteins such as caspases, contributing to the survival of melanoma cells and promoting tumor growth [88]. PAF also promotes angiogenesis in melanoma by stimulating the expression of VEGF and FGF2; furthermore, PAF enhances the activity of MMPs in melanoma cells, promoting the invasion of endothelial cells into surrounding tissues, thereby promoting angiogenesis [89]. PAF has immunosuppressive effects that can impair the anti-tumor immune response in melanoma; it inhibits the function of immune cells such as natural killer (NK) cells, T cells, and dendritic cells, thereby reducing their ability to recognize and eliminate tumor cells [90]. PAF also promotes the production of immunosuppressive cytokines such as IL-10 and TGF-β while inhibiting the release of pro-inflammatory cytokines such as TNFα and interferon-gamma, allowing melanoma cells to evade immune surveillance and promoting tumor progression [90]. Systemic delivery of PAR-1 small interfering RNA (siRNA) incorporated into neutral liposomes inhibited the growth of melanoma in vivo [91]. Additionally, tumor samples from PAR-1 siRNA-treated mice showed a decrease in blood vessel density and VEGF, IL-8, and MMP-2 expression levels in comparison to control animals.

Figure 1 summarizes the molecules involved in angiogenesis in melanoma.

## 4. Cells in the TME

### 4.1. Cancer-Associated Fibroblasts (CAFs)

CAFs originate from various cell types, including resident fibroblasts, bone marrow-derived mesenchymal stem cells, and endothelial cells undergoing EndMT [92]. In response to signals from melanoma cells and other components of the TME, such as inflammatory cytokines, growth factors, and ECM proteins, fibroblasts become CAFs [92]. Activated CAFs exhibit distinct phenotypic and functional changes, including increased expression of α-smooth muscle actin (α-SMA), fibroblast activation protein (FAP), and ECM-modifying enzymes such as MMPs [93]. CAFs play a critical role in promoting melanoma progression through various mechanisms: (1) CAFs contribute to the remodeling of the ECM by secreting ECM components such as collagen, fibronectin, and hyaluronic acid, and this altered ECM architecture facilitates tumor cell invasion and metastasis. (2) CAFs secrete growth factors such as FGFs, hepatocyte growth factor (HGF), and insulin-like growth factors (IGFs), which promote melanoma cell proliferation, survival, and angiogenesis. (3) CAFs can modulate immune responses within the TME by secreting immunosuppressive factors such as IL-6, IL-10, and TGF-β, as well as by promoting the recruitment and activation of immunosuppressive immune cell populations such as regulatory T cells (Tregs) and myeloid-derived suppressor cells (MDSCs). (4) CAFs contribute to therapy resistance in melanoma by secreting factors that promote tumor cell survival and by inducing changes in the ECM that limit drug penetration and efficacy [94,95,96]. CAFs produced and secerned several cytokines and chemokines, including IL-1α, IL-1β, IL-6, IL-8, and CXCL10, but also overexpressed programmed death ligand-1 and/or -2 (PD-L1/PD-L2), confirming that CAFs promotes invasion of melanoma with the release of different molecules, which may act both alone and in synergic combination [97].

### 4.2. Mast Cells

Invasive melanomas have a higher mast cell density compared to benign nevi and in situ melanoma (MIS) [98]. A strong association between the number of microvessels, tumor cells positive for FGF-2, mast cell count, and tumor growth has been established in human melanoma [99]. Moreover, mast cells and vasculature have a close spatial connection that characterizes the mast cell spatial distribution in melanoma [100]. In samples of cutaneous malignant melanomas, mast cells present in the dermis immunoexpressed VEGF, mast cell density, and microvascular density have a prognostic significance, with patients with high values having a shorter survival rate [101]. The distribution of mast cells in the intratumoral and peritumoral microenvironment of 51 MM patients was examined [102]. In stages 1, 2, and 3 of MM, the mean ± standard deviation (SD) of intratumoral mast cells were 9.4 ± 4.2, 10.8 ± 5.1, and 2.1 ± 2.3, with a *p* = 0.000. Additionally, mast cells in phases 1, 2, and 3 had mean ± values of 13.4 ± 2.4, 16.6 ± 2.4, and 8.2 ± 4.6 in the peritumoral microenvironment. Therefore, the authors reported a substantial direct correlation between the tumor’s depth and the decreased presence of tumor-infiltrating lymphocytes (TILs) and mast cell distribution. This finding raises the possibility that infiltrating mast cells and lymphocytes have an inhibitory effect, as demonstrated previously. A recent review article underscored the potential role of mast cells in promoting the growth and invasiveness of melanoma [103].

### 4.3. Melanoma-Associated Macrophages (MAs)

MAs play a critical role in shaping the TME and influencing disease progression, exhibiting plasticity, and adopting distinct functional phenotypes in response to microenvironmental signals. In melanoma, indeed, macrophages primarily exist in two polarized states: the classically activated (M1) phenotype and the alternatively activated (M2) phenotype [104]. M1 macrophages are characterized by pro-inflammatory and anti-tumor activities, producing pro-inflammatory cytokines such as IL-12 and TNF-α and promote cytotoxic T cell responses against melanoma cells, while M2 macrophages display anti-inflammatory and pro-tumor activities, secreting immunosuppressive cytokines such as IL-10 and TGF-β and contributing to tumor progression by promoting angiogenesis, tissue remodeling, and immune evasion [105]. Targeting macrophages represents a promising therapeutic strategy for melanoma treatment through the following processes: (1) M1 repolarization of macrophages, shifting the balance from M2-like pro-tumor macrophages to M1-like anti-tumor macrophages through immunomodulatory agents or cytokine therapy; (2) depletion of macrophages, eliminating tumor-promoting macrophages using targeted antibodies or small molecules that selectively target macrophage-specific markers or signaling pathways; (3) inhibition of macrophage recruitment, by targeting chemokine receptors or ligands involved in macrophage recruitment, such as CCL2/CCR2 axis inhibitors [106,107]. Figure 2 presents two examples of melanoma-associated macrophages, which are highlighted by CD163, a lineage marker.

### 4.4. External Conditions

In addition to intrinsic conditions, it is important to underline that it has been shown that melanoma cells can also be modified in their biological behavior by external conditions and factors. In particular, TME exhibits a lower pH (pH 5.5–7.4) compared to normal tissue due to metabolic alterations, including upregulation of glycolysis (even in the presence of oxygen, Warburg effect), highly active pentose phosphate pathway, and glutaminolysis [108,109,110]; furthermore, hypoxia, characteristic of solid tumors, further contributes to acidification via the hypoxia-inducible factor (HIF) pathway, that promotes glycolytic phenotype and lactate production, further contributing to acidification [111]. Finally, HIF activity is already increased in melanoma cells under normoxic conditions, potentially contributing to the glycolytic phenotype and acidification [112]. Cancer cells upregulate transporters and channels to maintain intracellular pH and export excess acid, contributing to extracellular acidification; these include monocarboxylate transporters, Na+/H+ exchangers, carbonic anhydrases, vacuolar-type H+-ATPases, and aquaporins [113]. Extracellular acidosis leads to the development of a more aggressive phenotype of melanoma cells, promoting metastases and enhancing invasiveness. Acid-resistant phenotypes are selected under acidic conditions, and when returned to physiological pH, these cells maintain higher invasive and motility potential [114].

## 5. Data from Next-Generation Sequencing (NGS) and Single-Cell RNA Sequencing (scRNAseq)

In recent years, Next-Generation Sequencing (NGS) has allowed us to make progress in clinical applied research, particularly in the context of melanoma, with particular molecular techniques that have contributed strongly to clarifying and discovering new important characteristics of the TME [115,116]. Among these, single-cell RNA sequencing has allowed us to characterize in detail not only the different cell populations present in the TME but also their functional activation/deactivation status, allowing us to examine the dynamism of the TME in greater detail [117]. In this regard, melanoma cells exhibit intrinsic plasticity and can rapidly adapt to various extracellular stresses (such as drug treatments, inflammation, and nutrient and oxygen deprivation), transitioning between differentiated and dedifferentiated states; this transition involves reversible switches to multiple discrete phenotypes within a tumor, characterized by differential expression of melanocytic markers such as MITF and SOX10 (most expressed in differentiated state) and AXL and SOX9 (most expressed in dedifferentiated state). Indeed, differentiated states are proliferative with high MITF expression, while dedifferentiated/or undifferentiated states are invasive and mesenchymal-like with low MITF expression. Furthermore, multiple intermediate states with unique transcriptional signatures have been identified through scRNA sequencing, and cells can transition bidirectionally between these states [118,119]. On the other hand, the concept of intra-tumoral heterogeneity (ITH) is also related to stromal and immune cells, such as myeloid cells and lymphoid cells that show high heterogeneity, with distinct functional roles based on polarization. For example, it is now clear that in TME (beyond T and B cells with some activation states such as dysfunctional, exhausted, activated), M1 and M2 macrophages, N1 and N2 neutrophils, M1 and M2 myeloid-derived suppressor cells (MDSC), and dendritic cells are also present. While alternatively activated myeloid cells are anti-inflammatory and promote tumor growth, classically activated neutrophils, macrophages, and MDSCs exhibit pro-inflammatory and anti-tumor properties [110,111,112,113,114,115,116,117,118,119,120,121,122]. Finally, CAFs undergo phenotypic switching, further adding to the complexity of ITH [123]. In this regard, a recent paper investigated the role of CAFs in cutaneous melanoma with the aim of identifying potential biomarkers and therapeutic targets associated with CAFs. The authors employed a Weighted Gene Co-expression Network Analysis (WGCNA) to identify CAF signature genes in melanoma and conducted a bioinformatics analysis to assess the CAF risk score and its association with tumor progression. Furthermore, scRNAseq and spatial transcriptome analysis were also employed to verify the expression and distribution patterns of signature genes. The most important results were the identification of FBLN1 and COL5A1 as crucial CAF signature genes and the development of a CAF risk score based on these genes. Furthermore, bioinformatics analysis revealed that high CAF risk scores were significantly associated with pathways related to tumor progression and that CAF risk scores were negatively correlated with clinical prognosis, immunotherapy response, and tumor mutational burden in CM patients. FBLN1 and COL5A1 were confirmed as CAF-specific biomarkers in CM through multi-omics analysis and experimental validation, and finally, drugs such as Mifepristone and Dexamethasone were identified as potential anti-CAF drugs based on these targets [94].

Regarding the therapeutic ways, ITH is a critical determinant of treatment response and resistance, leading to the selection and outgrowth of drug-resistant cancer cells, and some papers have demonstrated that increased ITH is associated with resistance to targeted therapies and immunotherapies, affecting treatment outcomes [124]. Challenges remain in the widespread adoption of scRNA-seq due to cost, technical complexity, and data analysis requirements, so future research should focus on leveraging scRNA-seq to dissect the heterogeneity of melanoma cell states, identify therapeutic targets, and overcome treatment resistance.

## 6. Anti-Angiogenesis in Melanoma

Patients with melanoma who were diagnosed at an early stage could be cured by surgical removal. Clinical therapeutic options include chemotherapy, immunotherapy, and other targeted therapies, such as anti-angiogenic agents. The main anti-angiogenic drug for metastatic melanoma is bevacizumab, a humanized VEGF monoclonal antibody [125]. A recent meta-analysis was conducted to evaluate the efficacy and safety of Bevacizumab in combination with other therapeutic regimens in the treatment of melanoma. The researchers included both randomized controlled trials (RCTs) and non-comparative clinical studies to investigate the use of bevacizumab in combination with chemotherapy (fotemustine, dacarbazine, carboplatin/paclitaxel, and temozolomide), targeted therapies (imatinib, everolimus, sorafenib, erlotinib, and temsirolimus) and interferon-gamma (INF-γ). From the search strategy, the authors included a total of 20 trials, including 5 RCTs and 15 non-comparative clinical studies. As a primary outcome, the pooled objective response rate was found to be 15.8%, and Bevacizumab combined with carboplatin/paclitaxel significantly improved overall survival compared to carboplatin/paclitaxel alone. The most important adverse events were fatigue, nausea, leukopenia, thrombocytopenia, and neutropenia, with the pooled incidence of hypertension across all bevacizumab arms at 32.4% [126].

Ramucirumab, a fully humanized monoclonal antibody targeting the VEGFR-2, can be employed alone or in combination with dacarbazine in chemotherapy-naïve patients with metastatic melanoma (MM) [127]. In real life, the most important study about Ramucirumab was conducted where patients were randomly assigned to receive either ramucirumab in combination with dacarbazine (Arm A) or ramucirumab alone (Arm B) every three weeks, with a dose of 10 mg/kg for Ramucirumab a dose of 1000 mg/m^2^ for dacarbazine. The primary endpoint was progression-free survival (PFS), with secondary endpoints including overall survival (OS), overall response rate, and safety assessments. A total of 106 patients were randomized, with 102 receiving the assigned treatment (52 in Arm A and 50 in Arm B), and baseline characteristics were similar between the two arms. The median PFS was 2.6 months in Arm A and 1.7 months in Arm B. The 6-month PFS rates were 30.7% and 17.9%, and the 12-month PFS rates were 23.7% and 15.6%, respectively. Regarding overall response (OR) in Arm A, 17.3% of patients had a partial response (PR), and 36.5% had stable disease (SD), while in Arm B, PR was observed in 4.0% of patients, and 42.0% had SD. Median OS was 8.7 months in Arm A and 11.1 months in Arm B. Both treatment regimens were well-tolerated, with limited Grade 3/4 toxicities reported. These findings suggested that ramucirumab, either alone or in combination with dacarbazine, could be a promising therapeutic option for chemotherapy-naïve patients with metastatic melanoma, warranting further investigation in larger clinical trials [127].

Aflibercept (VEGF Trap), a selective humanized IgG1 monoclonal antibody that can block the interaction of VEGF with VEGFR1 and VEGFR2, has been used in combination with pembrolizumab in melanoma [128]. Recently, a study investigated the combination of ziv-aflibercept anti-angiogenic therapy with pembrolizumab in patients with advanced melanoma resistant to anti-PD-1 treatment. Ten patients with advanced PD-1-resistant melanoma were enrolled and received a combination of ziv-aflibercept (at doses ranging from 2–4 mg/kg) plus pembrolizumab (at a dose of 2 mg/kg), administered intravenously every 2 weeks. Like data analysis, baseline and on-treatment plasma and peripheral blood mononuclear cell (PBMC) samples were analyzed using multiplex protein assay and mass cytometry. Two patients (20%) achieved a partial response to the combination therapy, while two patients (20%) experienced stable disease as the best response. Notably, the two responders had mucosal melanoma, while both patients with stable disease had ocular melanoma. The combination therapy demonstrated acceptable safety despite the occurrence of adverse events. Furthermore, changes in plasma analytes such as PDGF and PD-L1 were observed, indicating potential alterations in myeloid cell function.

Higher levels of circulating CXCL10 in non-responding patients could reflect pro-tumor activity. Specific subsets of γδ T cells were associated with poor clinical outcomes, suggesting impaired γδ T-cell function in non-responding patients [129].

Sorafenib, a raf kinase inhibitor, which can also inhibit the tyrosine kinase activity of various receptors, including VEGFR-2-3, PDGF-R, and KIT, has been used in patients with metastatic uveal melanoma [130]. In a study, sorafenib inhibited multiple signaling pathways implicated in melanoma progression, including c-Kit, PDGFR, VEGFR, and RAF, demonstrating efficacy in both in vitro and in vivo models [131].

In melanoma treatment, Lenvatinib, an oral multiple tyrosine kinase inhibitor, has been employed to modulate VEGFR1-3, FGFR1-4, PDGFR, and KIT [132]. In a study, the researchers employed various preclinical models, including subcutaneous and left ventricle melanoma models, to evaluate the effects of the treatment combinations. They utilized cytokine/chemokine profiling and flow cytometry to analyze changes in signaling and immune-infiltrating populations within the TME. Additionally, an in vitro blood–brain barrier (BBB) model was utilized to study the effects of treatment on endothelial integrity and leukocyte transmigration, particularly relevant for intracranial disease. The study demonstrated that dual treatment with anti-PD-1 and either anti-VEGF or Lenvatinib improved survival and reduced tumor growth in systemic melanoma models. Importantly, both combination therapies led to increased Th1 cytokine/chemokine signaling within the TME, indicating enhanced immune activation. However, the effects on immune cell populations differed between the two combinations, with Lenvatinib decreasing tumor-associated macrophages while increasing plasmacytoid dendritic cells (DCs) early in treatment.

Moreover, both Lenvatinib and anti-VEGF reduced intratumoral blood vessels, suggesting effective inhibition of angiogenesis. Interestingly, while anti-VEGF promoted endothelial stabilization in the in vitro BBB model, Lenvatinib did not exhibit the same effect. However, both treatments facilitated leukocyte transmigration, indicating the potential for enhanced immune cell infiltration into the brain metastatic site [133].

Sunitinib, an oral multi-kinase inhibitor, has been used in melanoma and targets VEGFRs, KIT, and other receptors [134,135]. In a recent work, the researchers focused on the combination of JQ1, a BET inhibitor, with Sunitinib, employing a comprehensive drug synergy screening approach, testing the combination of JQ1 with 240 antitumor drugs from the FDA-approved library. Following the identification of sunitinib as a synergistic partner, further experiments were conducted to elucidate the molecular mechanisms underlying the observed synergy. This included assessing apoptosis induction, cell cycle arrest, and alterations in gene expression patterns in melanoma cells treated with combination therapy. Additionally, in vivo xenograft assays were performed to validate the therapeutic efficacy of the combination in suppressing melanoma growth. The study revealed that the combination of JQ1 with sunitinib synergistically induced apoptosis and cell cycle arrest in melanoma cells. Mechanistically, BET inhibitors sensitized melanoma cells to sunitinib by inhibiting the expression of growth differentiation factor 15 (GDF15), a protein associated with tumor progression and resistance to therapy. Importantly, the transcriptional regulation of GDF15 was found to be directly influenced by BRD4, a key component of the BET protein complex, or indirectly via the BRD4/IL6/STAT3 signaling axis [136].

## 7. Conclusions

Anti-angiogenic therapy is less effective, and patients with melanoma invariably develop treatment resistance, especially if using bevacizumab monotherapy. The intrinsic and acquired limitations of anti-angiogenic drugs that contribute to their therapeutic benefit include the following: The tumor’s resistance to antiangiogenic therapy; the selection of resistant clones and the activation of alternative mechanisms that activate angiogenesis even when the drug’s target remains inhibited; therapy-induced reduction in oxygen levels within the tumor; accumulation of infiltrating cancer stem cells; activation of pro-invasive mechanisms and increased dissemination and metastatic potential; normalization of tumor blood vessels; recruitment of inflammatory and immature myeloid cells; alternative mechanisms of tumor vessel formation; and genomic instability of tumor endothelial cells. The idea that vascular co-option can mediate both innate and acquired resistance to anti-angiogenic therapy is supported by preclinical investigations. Co-option may also help to explain why anti-angiogenic cancer therapies are ineffective.

The idea and approaches of anti-angiogenic therapies need to be thoroughly revised and re-evaluated in this light. To overcome resistance, pharmacokinetic and pharmacodynamic data must be used to rationally combine anti-angiogenic drugs. Determining the ideal duration and timing of anti-VEGF agents is also crucial.

Recent clinical trials have investigated the efficacy of the association between anti-angiogenic agents and immune checkpoint inhibitors in advanced melanoma. Finally, the identification of specific predictive biomarkers remains an important endpoint, even if biomarkers that are predictive of anti-VEGF therapy may be specific to different tissues and tumor subtypes. The identification of reliable biomarkers of treatment response and the use of anti-angiogenic agents with conventional chemotherapy and immunotherapy may improve the care of individuals with cancers.

## Figures and Tables

**Figure 1 cancers-16-01794-f001:**
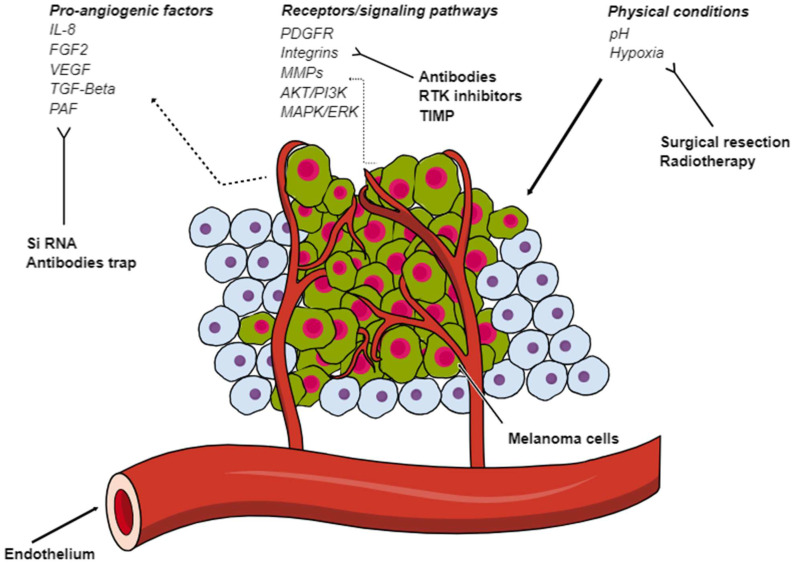
Overview of the molecules involved in angiogenesis in melanoma.

**Figure 2 cancers-16-01794-f002:**
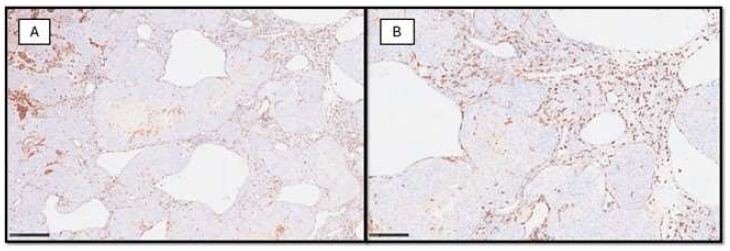
(**A**) Photomicrographs of an example of invasive malignant melanoma with a high density of macrophages (highlighted by brown chromogen) within the tumor sheets (immunohistochemistry for CD163, original magnification 4×). (**B**) Scanning magnification of CD163+ macrophages associated with malignant melanoma (immunohistochemistry for CD163, original magnification 10×).

## Data Availability

Data are contained in the manuscript.

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
