# Peer review of "Angiogenesis Still Plays a Crucial Role in Human Melanoma Progression"

_cancers, 2024, doi:10.3390/cancers16101794_

Round 1
Reviewer 1 Report
Comments and Suggestions for Authors
The review is dedicated to the angiogenesis in melanoma and summarizes factors that mediate angiogenesis in melanoma progression. The therapeutic options associated with targeting angiogenesis are touched upon in passing.
Overall, the article is written in good language and is easy to read.
There are some comments on the content
the authors' conclusions are confirmed by the literature sources used
selective checking of references confirms the adequacy of the interpretation of data from other articles.
My points on the article:
1. I recommend that you add more new links (from 2019), even if they repeat the old ones in some way. I think it will be more solid this way.
2. Point out in the introduction that lymphatic vessel density is associated with possible metastasis of primary melanomas (e.g., use reference 42).
3. Melanomas are primary and metastatic - try to indicate what type of melanoma was meant for the references (for example, in references 21, 22 metastatic melanomas are meant).
4. You divided the review into chapters: factors, integrins and MMPs, cells. I recommend adding another section about external conditions - melanoma is characterized by hypoxia and low pH, which can stimulate angiogenesis. Check out these articles - doi: 10.1111/pcmr.12495, 10.3390/ijms24054430. Adding this part will complete the review.
5. MAST cells - remove p = 0.000 values. Discuss newer articles - like this one doi: 10.1097/CMR.0000000000000932.
6. Secreted factors can be induced by activation of different receptors, for example, activation of nAChRs can cause the secretion of VEGF, note this, also receptors can form a functional network, thus, angiogenic factors can also mediate the progression of melanomas, and vice versa, different factors in the progression of melanomas can mediate angiogenesis Add a sentence about this to the introduction.
7. I recommend making a graph in which you combine the following things:
- what secreted factors enhance angiogenesis,
- what receptors are activated during angiogenesis,
- what factors are secreted by non-melanoma cells for angiogenesis,
- what can be done to influence receptors (RTK inhibitors, Ab), secreted factors (trap antibodies), or pro-angiogenic conditions of the tumor (surgery).
I made a messy draft (supplement 1), maybe you can get something from it. above is a picture of secreted factors,
blue - 3 columns: pro-angiogenic secreted factors, pro-angiogenic receptors/signaling pathways in melanoma cells, pro-angiogenic physical conditions of melanomas.
Finally green - which can be used for the treatment of melanomas. To destroy physical conditions - surgery and radio, for factors - At-traps or antisense RNA (as for PAF), maybe immunotherapy? for receptors and pathways - inhibitors and antibodies.
Little things:
- Add doi or link to all articles (doi is missing for example in ref 22, 54,75, reference 22 is difficult to find at all)
- Part about FGF2 - use correlation instead of ratio
- "VEGF121 and VEGF165" - replace with "isoforms VEGF121 and VEGF165"
- PAF - indicate that it is also called AGEPC.
Overall, the article is interesting, I would recommend it for publication after revision.

Author Response
Reviewer n’1: The review is dedicated to the angiogenesis in melanoma and summarizes factors that mediate angiogenesis in melanoma progression. The therapeutic options associated with targeting angiogenesis are touched upon in passing.
Overall, the article is written in good language and is easy to read.
There are some comments on the content the authors' conclusions are confirmed by the literature sources used
selective checking of references confirms the adequacy of the interpretation of data from other articles.
Answer n’1: First of all, thank you very much dear Reviewer n’1 for these wonderful words. We will try to do our best to improve the quality of our manuscript.
Reviewer n’1: 1. I recommend that you add more new links (from 2019), even if they repeat the old ones in some way. I think it will be more solid this way.
Answer n’2: Thank you very much for this useful tip dear Reviewer n’1. We have added some new references.
Reviewer n’1: 2. Point out in the introduction that lymphatic vessel density is associated with possible metastasis of primary melanomas (e.g., use reference 42).
Answer n’3: Dear Reviewer n’1, thank you very much. We have added two sentences about this aspect in Introduction.
Reviewer n’1: Melanomas are primary and metastatic - try to indicate what type of melanoma was meant for the references (for example, in references 21, 22 metastatic melanomas are meant).
Answer n’4: Ok dear Reviewer n’1. We have specified the “primary” or “metastatic” melanoma when it was necessary to understand deeply the study discussed. Thank you.
Reviewer n’1: 4. You divided the review into chapters: factors, integrins and MMPs, cells. I recommend adding another section about external conditions - melanoma is characterized by hypoxia and low pH, which can stimulate angiogenesis. Check out these articles - doi: 10.1111/pcmr.12495, 10.3390/ijms24054430. Adding this part will complete the review.
Answer n’5: Dear Reviewer n’1, thank you very much. We have added a section titled “external conditions” in which we have summarized the most important and update knowledges about hypoxia and low PH. Thank you.
Reviewer n’1: 5. MAST cells - remove p = 0.000 values. Discuss newer articles - like this one doi: 10.1097/CMR.0000000000000932.
Answer n’6: Thank you very much dear Reviewer n’1. We have added and briefly discussed this article but we are unable to read it because non open access. Thank you.
Reviewer n’1: 6. Secreted factors can be induced by activation of different receptors, for example, activation of nAChRs can cause the secretion of VEGF, note this, also receptors can form a functional network, thus, angiogenic factors can also mediate the progression of melanomas, and vice versa, different factors in the progression of melanomas can mediate angiogenesis Add a sentence about this to the introduction.
Answer n’7: Done. Thank you.
Reviewer n’1: 7. I recommend making a graph in which you combine the following things:
- what secreted factors enhance angiogenesis,
- what receptors are activated during angiogenesis,
- what factors are secreted by non-melanoma cells for angiogenesis,
- what can be done to influence receptors (RTK inhibitors, Ab), secreted factors (trap antibodies), or pro-angiogenic conditions of the tumor (surgery).
I made a messy draft (supplement 1), maybe you can get something from it. above is a picture of secreted factors,
blue - 3 columns: pro-angiogenic secreted factors, pro-angiogenic receptors/signaling pathways in melanoma cells, pro-angiogenic physical conditions of melanomas.
Finally green - which can be used for the treatment of melanomas. To destroy physical conditions - surgery and radio, for factors - At-traps or antisense RNA (as for PAF), maybe immunotherapy? for receptors and pathways - inhibitors and antibodies.
Answer n’8: Dear Reviewer n’1, we made the angiogenesis picture. Thank you very much for your useful guidance.
Reviewer n’1: Little things:
- Add doi or link to all articles (doi is missing for example in ref 22, 54,75, reference 22 is difficult to find at all)
Answer n’9: Thank you very much dear Reviewer n’1. MDPI style guide advice the authors to not report the DOI for the articles. Thank again.
Reviewer n’1: Little things:
- Add doi or link to all articles (doi is missing for example in ref 22, 54,75, reference 22 is difficult to find at all)
- Part about FGF2 - use correlation instead of ratio
- "VEGF121 and VEGF165" - replace with "isoforms VEGF121 and VEGF165"
- PAF - indicate that it is also called AGEPC.
Overall, the article is interesting, I would recommend it for publication after revision.
Answer n’10: Done. Thank you very much.
Reviewer 2 Report
Comments and Suggestions for Authors
There are a few issues that must be researched again, for example line 42-44. According to some information, it appears to me that the mortality rate of melanoma has actually decreased in the past few years, probably due to a) improved early diagnosis and b) the effect of immune checkpoint inhibitors in therapy.
Next, single-cel RNA sequencing has played a massive role in understanding human cancers, including melanoma, but its only mentioned a single time in the entire review. I would recommend having a look at the most recent, and most important scRNA-Seq studies and include findings specifically on the presence of endothelial cells and vascularization, if there are.
Otherwise, i find most of the "chapters" on growth factors, especially PlGF, a stub... not really having a lot of substance or any novelty findings. Some of the references in this section are also quite old, between 20 and 30 years old; have there not been any novel insights? If not, why bother... just state (nothing new" and elaborate other chapters instead. Its true that the high time of growth factor receptor research may be over... but is the chapter closed?
Concerning the other chapters/subchapters, they also largely summarize old news. Many of the things mentioned here e.g., about MMPs have made it ibto textbooks already, I really dont need to read about them in a review ublished in 2024.
Maybe the most novel and up-to-date chapters are those on the tumor microenvironment, and the cell types contained in it. But then, I would also expect a few up-to-date references from newer technologies such as spatial expression analyses, and single-cell RNA sequencing. This is only sporadically the case, and there are now a mix of very old and recent references in this 2nd half of the manuscript. Especially concerning CAFs, there is an explosion of new data coming out every months now, with all the different CAF subtypes and their diverse roles and functions, especially on activating or modulating the immune system. But what is said about the immune-modulatory role of CAFS here (267 - 273) is also not very novel, nor is it very detailed and informative. Its more or less superficial information.
For me personally, the most novel and informative content in this review is in the very last chapters, such as on melanoma-associated macrophages, but even here, we mostly hear about the "usual suspects" as key molecules mentioned in the text. Not a lot of novelty either, I think.
Last chapter: antiangiogenesis therapies and drugs, its barely 1/2 a page, and again, its not a complete listing of the most relevant innovative and experimental therapies or clinical trials, one could eily write an entire review on this topic alone which would be more interesting than the current text (and this chapter).
In summary, I feel that this review is just scratching the surface in many areas without going into the more interesting and novel mechanistic or functional insights. Instead of covering everything on the surface, I would recommend the authors to focus on some key areas and then really explore and elaborate this field. That would be much more rewarding than writing a textbook-like chapter with little novelty.
Comments on the Quality of English Language
there are some grammatical issues but nothing that couldnt be fixed with a good online editor function (or grammarly etc).
Author Response
Reviewer n’2: There are a few issues that must be researched again, for example line 42-44. According to some information, it appears to me that the mortality rate of melanoma has actually decreased in the past few years, probably due to a) improved early diagnosis and b) the effect of immune checkpoint inhibitors in therapy.
Answer n’1: First of all, thank you very much dear Reviewer n’2 and sorry for this mistake. We have corrected this one according your precious suggestions. Thanks a lot.
Reviewer n’2: Next, single-cel RNA sequencing has played a massive role in understanding human cancers, including melanoma, but its only mentioned a single time in the entire review. I would recommend having a look at the most recent, and most important scRNA-Seq studies and include findings specifically on the presence of endothelial cells and vascularization, if there are.
Answer n’2: Thank you very much dear Reviewer n’2. We have added a detailed paragraph about the most update informations became available from NGS and scRNA sequencing techniques. We hope that now it’s fine.
Reviewer n’2: Otherwise, i find most of the "chapters" on growth factors, especially PlGF, a stub... not really having a lot of substance or any novelty findings. Some of the references in this section are also quite old, between 20 and 30 years old; have there not been any novel insights? If not, why bother... just state (nothing new" and elaborate other chapters instead. Its true that the high time of growth factor receptor research may be over... but is the chapter closed?
Concerning the other chapters/subchapters, they also largely summarize old news. Many of the things mentioned here e.g., about MMPs have made it ibto textbooks already, I really dont need to read about them in a review ublished in 2024.
Answer n’3: First of all thank you very much dear Reviewer n’2 for your tips useful to improve our manuscript. So, firstly, we have update the PlGF section and FGF2 section because there were some other papers more recent in literature. Otherwise, the sections about VEGF and VEGFR are update with recent references. Furthermore, we have added a detailed paragraph about the “external conditions” that can be stimulate the angiogenesis in melanoma. Furthermore, we have added a long and detailed paragraph about the data from single cell RNA sequencing with the most update informations. We would like to maintain the more “ancient” informations about some mediators to give to readers a wide look to changes and nex informations about neoangiogenesis. Thanks a lot for everything.
Reviewer n’2: Maybe the most novel and up-to-date chapters are those on the tumor microenvironment, and the cell types contained in it. But then, I would also expect a few up-to-date references from newer technologies such as spatial expression analyses, and single-cell RNA sequencing. This is only sporadically the case, and there are now a mix of very old and recent references in this 2nd half of the manuscript. Especially concerning CAFs, there is an explosion of new data coming out every months now, with all the different CAF subtypes and their diverse roles and functions, especially on activating or modulating the immune system. But what is said about the immune-modulatory role of CAFS here (267 - 273) is also not very novel, nor is it very detailed and informative. Its more or less superficial information.
Answer n’4: Thank you very much dear Reviewer n’2. We have improved all sections about CAFs and now there are cited some important papers regarding this matter. We hope that now it’s better for you.
Reviewer n’2: For me personally, the most novel and informative content in this review is in the very last chapters, such as on melanoma-associated macrophages, but even here, we mostly hear about the "usual suspects" as key molecules mentioned in the text. Not a lot of novelty either, I think.
Answer n’5: Thank you very much. We have improved also this section with detailed informations about M1 and M2 polarized macrophages. We hope that now it’s better for you.
Reviewer n’2: Last chapter: antiangiogenesis therapies and drugs, its barely 1/2 a page, and again, its not a complete listing of the most relevant innovative and experimental therapies or clinical trials, one could eily write an entire review on this topic alone which would be more interesting than the current text (and this chapter).
Answer n’6: Dear Reviewer n’2, thanks again for these important suggestions. So, we have deeply modified also this section regarding angioangiogenesis therapies, with important and RECENT references regarding the most update therapeutic modalities. After this hard work, we hope that the manuscript is better for you. A warm greeting and thanks again.
Reviewer 3 Report
Comments and Suggestions for Authors
The manuscript did an extensive literature review and summarized results from studies investigating angiogenesis in melanoma, discussed potential uncertainties and tried to answer new questions about the relationship between neoangiogenesis and TM.
However, this manuscript is limited to summarizing the evidence from previous studies. Interpretation of the results and adding their perspectives/thoughts are lacking. For example, it was mentioned in Section 4.2 that “Therefore, the authors reported a substantial direct correlation between the tumor's depth and the decreased presence of tumor-infiltrating lymphocytes (TILs) and mast cell distribution “. The authors didn’t present perspectives or critical analysis on why this finding raises the possibility that infiltrating mast cells and lymphocytes have an inhibitory effect.
This review mentioned lots of previous studies but does not appear to deeply criticize the studies it mentions, like discussing the bias, and the strength of previous findings.
The inclusion and exclusion criteria for previous studies are not well-defined. Not sure if this review effectively covered the current state of research. It would be beneficial to include a comprehensive search strategy.
It was mentioned in Section 6 line 353 that “Recent clinical trials have investigated the efficacy of the association 353 between anti-angiogenic agents and immune-checkpoint inhibitors in advanced melanoma.”. Please cite those research.
While the author mentioned the anti-angiogenic therapies in the conclusion, the discussion on the effectiveness or limitation of the therapy is lacking.
Author Response
Reviewer n'3:
However, this manuscript is limited to summarizing the evidence from previous studies. Interpretation of the results and adding their perspectives/thoughts are lacking. For example, it was mentioned in Section 4.2 that “Therefore, the authors reported a substantial direct correlation between the tumor's depth and the decreased presence of tumor-infiltrating lymphocytes (TILs) and mast cell distribution “. The authors didn’t present perspectives or critical analysis on why this finding raises the possibility that infiltrating mast cells and lymphocytes have an inhibitory effect.
This review mentioned lots of previous studies but does not appear to deeply criticize the studies it mentions, like discussing the bias, and the strength of previous findings.
The inclusion and exclusion criteria for previous studies are not well-defined. Not sure if this review effectively covered the current state of research. It would be beneficial to include a comprehensive search strategy.
Answer n'1: Dear Reviewer n'3, thank you very much for your useful tips. So, we have deeply modified the texture of our manuscript with important and up-dates about the all sections of manuscript. Furthermore, we have added a section about "External Conditions" that can influence the angiogenesis in melanoma such as low pH and hypoxia; furthermore, we have improved the section of TME, with a explanaitory figure and more details from Next Generation Sequencing (NGS) and single cell RNA sequencing (scRNA seq). Finally, we state that this is a narrative type review, in which we want to give to readers a panoramic view about the history and evolution of neoangiogenesis in melanoma. Thank you.
Reviewer n'3:
It was mentioned in Section 6 line 353 that “Recent clinical trials have investigated the efficacy of the association 353 between anti-angiogenic agents and immune-checkpoint inhibitors in advanced melanoma.”. Please cite those research.
While the author mentioned the anti-angiogenic therapies in the conclusion, the discussion on the effectiveness or limitation of the therapy is lacking.
Answer n'2: Dear Reviewer n'3, thank you very much. We have cited and deeply discussed the references about the employment of combination of anti-angiogenic therapy and ICIs. Furthermore, we have discussed the ineffectivness of anti-angiogenic therapy, giving a perspectives and a critical judgement.
Thank you for everything.
Round 2
Reviewer 1 Report
Comments and Suggestions for Authors
The authors answered my comments properly and significantly increased the volume of "therapeutic" part of the review.
Added parts are well-written, easy to follow.
Main authors claims are supported by the references. Selective checking of references shows the adequacy of the authors’ description of the literature data.
My only point is that some parts (page 11, refs 128, 130, 133) may be shortened, however, it is not mandatory.
Overall, the article is interesting, and I can recommend it for publication.
Reviewer 2 Report
Comments and Suggestions for Authors
I read through the newy added passages and the manuscript has definitely gained in relevance due to these new sections. They have also added a considerable number of new references and generally expanded the scope of the manuscript. I think the manuscript is acceptable now for publication. Of course, a few things cold still be added here and there but as a whole, its now a lot more informative and contains more novelty compared to the 1st version.
Comments on the Quality of English Language
english is fine, nothing that cannot be fixed in production of the manuscript
Reviewer 3 Report
Comments and Suggestions for Authors
All my comments have been addressed